# Advances and Challenges in the Pursuit of Disease-Modifying Osteoarthritis Drugs: A Review of 2010–2024 Clinical Trials

**DOI:** 10.3390/biomedicines13020355

**Published:** 2025-02-04

**Authors:** Mckenzie D. Brandt, Jason B. Malone, Thomas J. Kean

**Affiliations:** 1College of Medicine, University of Central Florida, Orlando, FL 32827, USA; mc417656@ucf.edu; 2Biionix Cluster, Department of Internal Medicine, College of Medicine, University of Central Florida, Orlando, FL 32827, USA; 3Department of Orthopedic Surgery, Nemours Children’s Health System, Orlando, FL 32827, USA; jason.malone@nemours.org

**Keywords:** osteoarthritis, DMOAD, cartilage, small molecule, clinical trial

## Abstract

**Background/Objectives**: Osteoarthritis (OA) is a highly prevalent, degenerative joint disease capable of causing severe pain and impaired mobility. Current treatments mitigate symptoms but do not cure the disease. The development of a disease-modifying osteoarthritis drug (DMOAD) could improve patient outcomes by slowing, halting, or reversing joint damage. Many DMOADs have progressed to clinical trials, but very few have made a significant impact, and none have been approved for clinical use. The purpose of this review is to present an update on the current status of DMOADs with a particular focus on results published since 2010. **Methods**: A comprehensive search was conducted within PubMed and ClinicalTrials.gov for novel DMOADs enrolled in phase II and III clinical trials between 1 January 2010 and 1 July 2024. **Results**: Eleven DMOAD candidates are reviewed and critically analyzed for their potential benefit in OA treatment—Lorecivivint (SM04690), TissueGene-C, Cindunistat (SD-6010), Sprifermin, UBX0101, TPX-100, GLPG1972/S201086, Lutikizumab (ABT-981), SAR113945, MIV-711, and LNA043—and relevant challenges to their development are discussed. **Conclusions**: Six DMOADs have demonstrated statistically significant evidence of a structural or symptomatic benefit without major safety concerns in phase II and III randomized controlled trials post-2010.

## 1. Introduction

Osteoarthritis (OA) is a degenerative joint disease which causes chronic joint pain and stiffness. Its global impact is substantial, with its prevalence exceeding 600 million in 2020 and projected to reach 1 billion by 2050 [1]. The prevalence of OA increases in adults over 40 and rises steeply with age [1]. Additional risk factors include obesity, prior joint injuries, and being of the female sex [2]. The symptomatic burden of OA can be debilitating, contributing to a decreased quality of life and heightened risk of cardiovascular disease, diabetes mellitus, and obesity [3]. As a leading cause of disability among adults, the annual health care expenditures related to OA management are approximately $65 billion in the US alone [4].

While previously described as the isolated “wear and tear” of articular cartilage, OA is now regarded as a complex and multifactorial process involving the entire joint, subchondral bone, and periarticular structures [5]. The onset of OA is initiated when the balance between chondrocyte anabolism and matrix catabolism is perturbed. In healthy joints, chondrocytes synthesize shock-absorbing extracellular matrix (ECM) components—such as type II collagen, hydrated proteoglycans, and glycosaminoglycans (GAGs) [6]—while degradative matrix metalloproteinases (MMPs) participate in matrix turnover. In OA, chronic increases in pro-inflammatory cytokines such as tumor necrosis factor α (TNF-α) and interleukin-1β (IL-1β), declining chondrocyte proliferation, and the upregulation of MMPs can collectively injure joint structures [6,7,8]. Thinning of the articular cartilage, subchondral bone remodeling, and trabecular thickening occur in response to injury and exacerbate the inflammatory response. Additionally, the avascular nature of articular joints is a barrier to healing [9].

The clinical management of OA is multifaceted and includes a variety of nonpharmacologic, pharmacologic, and surgical options. First-line therapy typically includes regular exercise, the maintenance of a healthy body weight, and physical therapy to improve the strength and mobility of the affected joint [10,11]. When these strategies do not adequately manage symptoms, oral or topical non-steroidal anti-inflammatory drugs (NSAIDs) are available to temporarily reduce pain and inflammation. Intra-articular (IA) injections with corticosteroids and hyaluronic acid can be used to delay surgical treatment but are not routinely used or recommended for long-term use due to safety concerns [12,13,14]. While some patients develop mild, non-progressive OA and gain adequate symptomatic relief from first-line therapies alone, the majority will progressively deteriorate [15]. The definitive treatment for advanced OA is joint replacement surgery. Although joint replacements are highly effective for most patients, they are not available for every joint affected by OA and have limited longevity. The invasive nature of the procedure poses a substantial risk of infection and is contraindicated in patients with certain medical conditions. They may also be inaccessible to patients without health insurance, due to their high cost, and are unfavorable among younger patients.

While current therapies mitigate symptoms, none address the underlying progression of OA. Discovery of disease-modifying osteoarthritis drugs (DMOADs) which slow, halt, or even reverse degeneration of the affected joint would represent a significant advance in treatment. These agents could improve patient outcomes when added to a comprehensive treatment plan. Many drug targets have been proposed, and several compounds have shown promise in pre-clinical studies and advanced to clinical trials, but none have been approved and marketed as DMOADs. The goal of this review is to provide an updated analysis on the current status of DMOADs and identify notable achievements and challenges in their development. We will discuss DMOADs studied in phase II or III randomized controlled trials completed since 2010 to narrow our focus to agents of recent or current interest.

## 2. Methods

This review focused on novel DMOADs which had completed at least one phase II or III clinical trial post-2010. A search was conducted in PubMed in July 2024 using the search terms (Clinical trial) AND (osteoarthritis) AND ((Disease-modifying) OR (DMOAD) OR (small molecule)). The results were limited to exclude articles published before 2010. Literature reviews, pre-clinical trials, and quality improvement studies were excluded. These results were supplemented with a search for clinical trials using the clinicaltrials.gov online repository. A search for “osteoarthritis” was conducted, and the results were limited to display phase II and III interventional trials with a primary completion date between 1 January 2010 and 1 July 2024.

In order to be eligible for inclusion, compounds must have been novel, possessed a mechanism of action which was scientifically capable of modifying disease, and have completed at least one phase II or III interventional OA clinical trial with a primary completion or termination date between 1 January 2010 and 1 July 2024. Trials without results published on clinicaltrials.gov (accessed on 1 July 2024) or a full-text English language article available in PubMed were excluded.

## 3. Results

### 3.1. Study Selection

Figure 1 illustrates the study selection process utilized for this review. The search in PubMed yielded 146 results. Ten articles described relevant clinical trials and were included in the review [16,17,18,19,20,21,22,23,24,25]. The search on Clinicaltrials.gov (accessed on 1 July 2024) yielded 632 results. An additional 23 trials met the inclusion criteria, and their corresponding publications (14) were included in the review [26,27,28,29,30,31,32,33,34,35,36,37,38,39].

In total, 24 publications were reviewed, and 41 clinical trials were referenced (Table 1 and Table 2). Eleven DMOAD candidates were described: Lorecivivint (SM04690), TissueGene-C, Cindunistat (SD-6010), Sprifermin, UBX0101, TPX-100, GLPG1972/ S201086, Lutikizumab (ABT-981), SAR113945, MIV-711, and LNA043.

### 3.2. Lorecivivint

Lorecivivint (SM04690) is a small-molecule inhibitor of the Wnt/β-catenin signaling pathway which plays a key role in human skeletal development and is tightly regulated in healthy joints. The upregulation and complete knockout of β-catenin induces OA in murine models [40]. In pre-clinical studies, SM04690 promoted chondrogenesis in human mesenchymal stem cells (MSCs) and increased the cartilage thickness in rodents with an induced cruciate ligament injury [41].

SM04690 entered a 24-week phase I clinical trial in 2014 (NCT02095548) and proved safe in 61 subjects with moderate-to-severe knee OA [16]. A 52-week phase IIa trial (NCT02536833) was initiated in 2015, including 455 subjects, and compared 0.03 mg, 0.07 mg, and 0.23 mg single-dose IA Lorecivivint injections to a placebo. The study utilized Western Ontario and McMaster University Osteoarthritis Index (WOMAC) pain scores as its primary measure and WOMAC function scores and the joint space width (JSW) as secondary measures but failed to demonstrate statistically significant reductions in any measure [26]. However, post hoc analyses revealed that when subjects with bilateral symptomatic knee OA were excluded, the 0.07 mg cohort displayed statistically significant improvements in the WOMAC pain scores, WOMAC function scores, and JSW [26]. The inclusion of patients with bilateral symptomatic knee OA was identified as a potential confounding factor as these patients may have difficulty differentiating the source of their pain.

The study was extended for an additional 24 weeks in a revised phase IIb trial in 2017 (NCT03122860) and enrolled 700 subjects across 74 US locations with greater pain in one knee (the target knee) with respect to the other. The 0.07 mg and 0.23 mg cohorts demonstrated statistically significant improvements in the WOMAC pain and function scores [27], but there were no significant differences in the JSW among any cohort. Multiple phase III trials have concluded within the past 2–3 years with results pending (NCT05603754, NCT03928184, NCT04385303, NCT04520607). The phase III STRIDES trial assessed a single IA injection of 0.07 mg Lorecivivint into the most painful knee (NCT05603754) and was the most recent phase III trial to be completed, ending in February 2024. Lorecivivint has also proved safe when used in tandem with a corticosteroid, triamcinolone acetonide (NCT04598542), indicating that it has potential for use in multi-drug therapies, if approved [17].

### 3.3. TissueGene-C

TissueGene-C (TG-C) is an IA gene/cell therapy which contains a 3:1 ratio of human allogeneic chondrocytes and irradiated GP2-293 cells which have been engineered to overexpress transforming growth factor β (TGF-β). TGF-β is an anabolic cytokine which induces ECM synthesis in healthy tissues [42] and promotes an anti-inflammatory macrophage phenotype and tissue repair post-injury [43].

TG-C was developed and initially tested in South Korea, where it proved safe in two 2008 phase I trials (NCT02341391, NCT00599248) and effective in three phase II trials between 2009 and 2014 (NCT01671072, NCT02341378, NCT01825811). A 52-week phase III trial was completed in 2015 and included 163 subjects with degenerative knee OA (NCT02072070). TG-C proved significantly efficacious in improving pain and function when compared with a placebo [28] and has since received marketing approval in Korea. While TG-C tended to improve the cartilage thickness and slow subchondral bone growth, these results were not statistically significant during the study period [28].

In its first US clinical trial in 2014 (phase II), a single IA injection of TG-C significantly reduced the magnetic resonance imaging (MRI)-measured cartilage damage, synovitis incidence, and pain scores in 102 subjects with knee OA when followed for 1 year (NCT01221441) [29,30]. There was no significant reduction in osteophyte formation or other bone marrow lesions and no increase in adverse events when compared with the placebo cohort. TG-C is currently enrolled in two phase III US trials (NCT03203330 and NCT03291470), which are expected to conclude in 2025 and 2026, respectively. The two studies will each analyze changes in the JSW and WOMAC pain scores in over 500 subjects over a 2-year period and include a 15-year follow-up period to monitor safety.

### 3.4. Cindunistat

Cindunistat (SD-6010) is a selective inhibitor of inducible nitric oxide synthase (iNOS), an enzyme which mediates the synthesis of nitric oxide (NO) and is upregulated in many inflammatory conditions. NO contributes to OA pathogenesis by placing oxidative stress on the joints, activating MMPs, inducing chondrocyte apoptosis, and acting as a pro-inflammatory mediator [44]. A canine induced injury model found that OA cartilage explants produced an increased amount of NO in culture due to the upregulation of iNOS in chondrocytes [45]. The same study found that dogs treated with an iNOS inhibitor had reduced osteophytosis, smaller cartilage lesions, and less synovial inflammation on histology [45].

A multicenter, phase II clinical trial completed in 2011 evaluated Cindunistat in 1457 subjects with knee OA at 183 locations (NCT00565812). Participants were recruited in 14 countries across North and South America, Australia, and Europe. All enrolled subjects were overweight or obese (BMI of 25–40), due to the correlation between obesity and elevated iNOS activity [46]. The study found that subjects who took 50 or 200 mg oral Cindunistat tablets once daily for two years demonstrated no significant reduction in medial tibiofemoral joint space narrowing (JSN) or pain [31]. A subgroup analysis of patients with milder structural OA found that subjects taking 50 mg Cindunistat had a significantly lower rate of JSN at week 48, but this result was not sustained. A phase III trial was submitted to clinicaltrials.gov in 2012 (NCT01438918) but was withdrawn before participants were enrolled.

### 3.5. Sprifermin

Sprifermin is a synthetic analog of human fibroblast growth factor-18 (FGF-18), an innate protein which contributes to cellular growth and tissue repair [6]. FGF-18 induces chondrocyte proliferation in vitro and mitigates ECM destruction by downregulating MMPs [47]. Pre-clinical studies have demonstrated that Sprifermin increases the chondrocyte quantity and hyaline cartilage production in a 3-D culture model [48].

Sprifermin proved safe in a phase I trial completed in 2010 (NCT01033994) and entered a phase II trial soon after. The FORWARD study (NCT01919164) was a phase II 5-year trial investigating the effects of IA Sprifermin and is the longest completed DMOAD trial [20]. The trial concluded in 2019 and included 549 subjects with knee OA at 13 sites across eight countries. At the 2-year primary outcome mark, patients who received 100 μg Sprifermin injections every 6 months experienced an average increase in the femorotibial joint cartilage thickness of 0.05 mm, a statistically significant difference from the placebo group [18,22]. The results were dose-dependent, with a 0.04 mm average increase in the cartilage thickness among participants who received 100 μg every 12 months, a 0.02 mm average increase for participants who received 30 μg every 6 months, and a 0.01 mm average increase for participants who received 30 μg every 12 months. Subjects in the placebo cohort had an average decrease in the femorotibial joint cartilage thickness of −0.07 mm. No cohort reported a statistically significant decrease in the WOMAC pain score.

Following the 2-year treatment period was an extended 3-year follow-up period. A total of 378 participants completed the full 5-year follow-up, and the statistically significant 0.05 mm increase in the femorotibial joint cartilage thickness among patients who received 100 μg Sprifermin injections every 6 months was sustained to year 5 [20,22]. No subjects in this cohort required knee replacement surgery by the end of year 5, compared to 4.6% of subjects in the placebo group. Mild to moderate AEs were reported but were deemed independent of the treatment with Sprifermin and were not significantly greater in the treatment vs. the placebo group [20,22]. A phase III Sprifermin trial has not yet been initiated.

### 3.6. UBX0101

UBX0101 belongs to a class of drugs called senolytic agents, which selectively target and lyse senescent cells. Cellular senescence is one of the hallmarks of aging and occurs when cells upregulate antiapoptotic pathways to arrest their development in a terminally differentiated state. These cells accumulate and secrete inflammatory cytokines, matrix-degrading proteases, and harmful reactive oxygen species, contributing to tissue damage over time [49,50]. Introducing senescent cells into the joints of heathy mice through an IA injection induced degenerative joint disease [51], reinforcing the hypothesis that their removal could slow the progression of many diseases.

UBX0101 blocks the interaction between p53 and murine double minute 2 (MDM2), which typically provides negative feedback inhibition for p53 [52]. In the absence of this interaction, p53 can proceed unregulated and induce apoptosis in senescent cells. In pre-clinical studies, IA UBX0101 successfully cleared senescent cells, mitigated ECM loss, and alleviated symptoms in mice with post-traumatic OA [53]. UBX0101 was generally well tolerated in two phase I US trials between 2018 and 2020 (NCT04229225, NCT03513016) [54] and entered a phase II trial in late 2019 (NCT04129944). The trial spanned 24 weeks and included 183 subjects with knee OA at 20 US locations. Participants received a single IA injection of 0, 0.5, 2, or 4 mg UBX0101 and were monitored for symptomatic improvement, but no significant reduction in the WOMAC pain scores was observed [38]. An ongoing observational study assessing the long-term safety of UBX0101 was terminated following the release of these results (NCT04349956).

### 3.7. TPX-100

TPX-100 is a synthetic peptide analog derived from matrix extracellular phosphoglycoprotein (MEPE), a glycoprotein that is abundantly expressed in healthy osteocytes but downregulated in OA [55]. TPX-100 was subject to a 12-month phase II clinical trial at a single US medical center from 2014 to 2016 (NCT01925261) to analyze its role in cartilage regeneration. In phase IIa of the study, 118 subjects received four once-weekly IA injections of TPX-100 at 50, 100, or 200 mg to evaluate its safety. All doses were found to be reasonably safe and well tolerated.

In phase IIb, 93 subjects received four additional doses of 200 mg TPX-100 and underwent bilateral knee MRI at baseline, 6 months, and 12 months. The primary endpoint for this study was a change in cartilage thickness, but there was no significant difference seen on MRI at 6 or 12 months. However, the WOMAC pain scores showed a significant decrease in TPX-100-treated knees vs. placebo-treated knees at both the 6- and 12-month timepoints. Following the conclusion of the trial, a retrospective MRI analysis of the femoral bone shape found that knees injected with TPX-100 had significantly fewer osteophytes at the 6- and 12-month MRI exams than those treated with a placebo [23]. A phase III trial for TPX-100 has not yet been initiated.

### 3.8. GLPG1972/S201086

GLPG1972/S201086 (GLPG1972) is a small-molecule inhibitor of ADAMTS-5, one of the primary proteases that contributes to ECM turnover in OA. In pre-clinical trials, the suppression of ADAMTS-5 reduced the degradation of aggrecan—the major proteoglycan of articular cartilage ECM—in a human chondrocyte model [56]. ADAMTS-5 expression was necessary for OA progression in murine models, with knockout mice failing to develop symptoms of the disease following joint destabilization surgery [57,58].

GLPG1972 was subject to a phase I clinical trial in 2017 which evaluated its safety when administered via oral tablets in 30 subjects with hip and/or knee OA (NCT03311009). GLPG1972 was generally well tolerated at all doses, with headaches being the most common treatment-related AE [24]. A phase II trial was conducted from 2018 to 2020, which evaluated GLPG1972’s dose-ranging efficacy in 932 subjects with knee OA at 44 US locations (NCT03595618). Participants were randomized into four cohorts and received once-daily oral tablets of 0, 75, 150, or 300 mg GLPG1972 for 52 weeks. At the end of the treatment period, there was no significant reduction in the medial femorotibial compartment cartilage loss (assessed by MRI) or WOMAC pain scores in patients treated with any dose of GLPG1972 vs. a placebo [33]. A phase III trial has not been initiated.

### 3.9. Lutikizumab

Interleukin-1 (IL-1) is an inflammatory cytokine which triggers the inflammatory cascade in response to cellular injury. In chronic injuries, such as OA, the IL-1 pathway becomes constitutively active and exacerbates tissue damage. Lutikizumab (ABT-981) is an anti-IL-1α/β immunoglobulin which aims to suppress this pathway. ABT-981 entered a phase I clinical trial in 2012 to assess its safety when administered as multiple subcutaneous (SC) injections in 36 subjects with mild-to-moderate knee OA (NCT01668511). ABT-981 was generally well tolerated, with the most common treatment-related AE being redness at the injection site. Pharmacodynamic analysis confirmed reduced IL-1α/β levels in serum [35].

A phase IIa trial was performed from 2014 to 2016 and evaluated the efficacy of ABT-981 in patients with synovitis-associated knee OA (NCT02087904). A total of 350 participants were randomized to receive 0, 25, 100, or 200 mg SC ABT-981 biweekly for 50 weeks. There was no significant reduction in either the WOMAC pain scores or MRI evidence of synovitis for any dose of ABT-981 at the end of the 50-week treatment period [34]. Discontinuations and AEs such as injection site reactions and neutropenia occurred more frequently in patients treated with ABT-981 than a placebo. A phase III trial has not been initiated.

### 3.10. SAR113945

SAR113945 is a small-molecule inhibitor of IκB kinase (IKK), which activates the nuclear factor kappaB (NF-κB) pathway in the presence of the inflammatory cytokines IL-1β and TNF-α. As arthritic joints become increasingly inflamed, the activation of the NF-κB pathway shifts chondrocytes to a degradative state by recruiting catabolic enzymes such as MMPs and ADAMTS-5 [59]. NF-κB activation regulates inflammation via a positive feedback loop, initiating a vicious cycle in which additional inflammatory cytokines are recruited to the active site.

IA SAR113945 proved generally well tolerated in patients with knee OA across three phase I clinical trials conducted in Japan and Germany from 2010 to 2012 (NCT01113333, NCT01463488, NCT01511549). The most common AEs were nasopharyngitis, joint pain and swelling, and difficulty walking. A 6-month phase IIa trial was initiated in Germany in 2012 (NCT01598415) and included 130 subjects with knee OA. Participants were treated with a single IA injection containing SAR113945 or a placebo and followed for 168 days. The trial did not meet its primary endpoint of significantly reducing the WOMAC pain scores [32], and a phase IIb or III trial has not been initiated.

### 3.11. MIV-711

MIV-711 is a novel inhibitor of cathepsin K, a protease which mediates bone resorption and is upregulated in arthritic joints. [60] In an induced-injury animal model, treatment with MIV-711 suppressed biomarkers of bone resorption and cartilage degradation and reversed subchondral bone loss [61].

MIV-711 was tested in a phase IIa clinical trial at six European sites in 2016 (NCT02705625). A total of 244 participants with primary knee OA were treated with 0, 100, or 200 mg of oral MIV-711 capsules once daily for 26 weeks. The trial was granted an extension later that year and 50 participants continued treatment with 200 mg MIV-711 for an additional 26 weeks (NCT03037489). While MIV-711 did not significantly reduce pain, an MRI analysis of the medial femoral joint revealed significantly reduced thinning of the articular cartilage in the 100 mg cohort and reduced subchondral bone growth in both the 100 mg and 200 mg cohorts [37]. Treatment with MIV-711 was not associated with an increase in AEs.

Like the Lorecivivint trials, post hoc analyses were performed and revealed that a subgroup of patients with unilateral knee pain treated with 100 mg MIV-711 did report significantly reduced WOMAC pain scores after 26 weeks of treatment [36]. Altered inclusion criteria and an extended analysis of the 100 mg dose may be beneficial moving forward. Medivir, the pharmaceutical company responsible for MIV-711’s development, launched a phase I safety trial in the US in 2018 but has not posted results or registered any further trials.

### 3.12. LNA043

LNA043 is a derivative of angiopoietin-like protein 3 (ANGPTL3), a protein produced by the liver to regulate lipoprotein synthesis, and a relative of ANGPTL2 which is expressed in chondrocytes [62]. In pre-clinical studies, LNA043 stimulated chondrocyte differentiation in MSCs and regenerated the hyaline cartilage matrix by binding to the chondrocyte fibronectin receptor integrin α5β1. It also upregulated anabolic markers such as lubricin while suppressing known mediators of OA progression, such as Wnt, alkaline phosphatase, and adipokine leptin [62].

LNA043 entered its first clinical trial in 2015 and proved to be safe when administered via IA injection in 28 OA patients scheduled for total knee replacement (NCT02491281). The compound is currently under further investigation in the phase II ONWARDS trial, a two-part, 10-year study analyzing the efficacy and safety of multiple IA injections of LNA043 to regenerate the articular surface. Phase IIa (NCT03275064) included 142 participants with articular cartilage lesions of the knee and concluded in 2022. There was significantly less volume loss in patients treated with four weekly IA LNA043 injections (20 mg) vs. a placebo, although no symptomatic benefit was observed [39]. The most common AE observed was swelling near the injection site, and no serious AEs were related to the treatment. Phase IIb (NCT04864392) will test LNA043 in 581 subjects with knee OA at 76 locations across 15 countries and is expected to conclude in 2027. An LNA043–Canakinumab combination therapy was also enrolled in a phase II trial in 2021 (NCT04814368) but has been terminated for reasons unspecified.

## 4. Discussion

### 4.1. Clinical Relevance of Novel DMOADs

In an aging population, the discovery of disease-modifying treatments for OA could drastically improve the quality of life for millions affected by the disease. While statistically significant improvements in joint pain or the structure are certainly promising, the clinical relevance of a marginal increase in one without the other remains unclear. An agent which solely improves the joint structure but does not mitigate or prevent OA symptoms may be pointless in a clinical setting because it would not improve patient outcomes. Likewise, an agent which improves pain and physical activity without conserving the joint structure may actually accelerate disease progression by increasing joint loading and mechanical stress [63]. In the following subsections, we will discuss the practicality of each drug class—small-molecule inhibitors, recombinant protein analogs, gene therapies, senolytics, and monoclonal antibodies—from a clinical standpoint.

#### 4.1.1. Small-Molecule Inhibitors

Small-molecule inhibitors are typically organic, low-molecular-weight molecules which bind proteins to inhibit their activity. They make attractive DMOAD candidates because they are relatively cheap and easy to produce, but they can lack specificity and produce off-target effects. Five small-molecule inhibitors have been evaluated as DMOADs in phase II or III clinical trials since 2010: Lorecivivint, MIV-711, Cindunistat, SAR113945, and GLPG1972. These agents are designed to slow or prevent OA progression by targeting inflammatory and catabolic mediators. While they may be clinically useful in treating early-stage OA, they do not provide an anabolic benefit and may be incapable of restoring native joint function in late-stage disease.

The only small-molecule inhibitor providing a symptomatic benefit was Lorecivivint, which has shown great promise in its ability to successfully alleviate pain but has not yet demonstrated any structural benefit. Lorecivivint is the only small-molecular compound which has progressed to a phase III clinical trial, and the results of the four most recent trials are pivotal to its development as a DMOAD. Alternatively, MIV-711 demonstrated a structural, but not symptomatic, benefit. It significantly reduced articular cartilage thinning and subchondral bone growth in patients with knee OA, and its oral formulation would offer great convenience. Cindunistat, SAR113945, and GLPG1972 did not improve either measure, and the latter two compounds were associated with inflammatory reactions and headaches, respectively.

#### 4.1.2. Recombinant Protein Analogs

In contrast to the anti-catabolic mechanisms of the small-molecule inhibitors, synthetic protein analogs often aim to restore anabolism in arthritic joints. These compounds are also relatively cheap and easy to produce, but the three identified here—Sprifermin, TPX-100, and LNA043—do not address the degradative processes involved in OA pathogenesis. These agents may be beneficial in treating either early- or late-stage OA and have had relatively positive results in clinical trials thus far.

Sprifermin and LNA043 improved the joint structure. A series of four IA Sprifermin injections increased the femorotibial joint cartilage thickness by 0.05 mm over a 2-year period, a result which was statistically significant but perhaps lacks clinical relevance in the absence of a symptomatic benefit. An increase of 0.05 mm is approximately 2.5% of the non-arthritic femorotibial joint cartilage thickness [64], a rather marginal increase by clinical standards. However, Sprifermin did safely modify the cartilage thickness in a dose-dependent manner, and these effects were sustained throughout the 5-year follow-up period. Further investigation is needed to determine whether a longer treatment period or an increased treatment frequency could provide a greater structural benefit. LNA043 appears to delay volume loss in articular cartilage lesions, but, like Sprifermin, does not mitigate pain. The results of the ongoing phase IIb ONWARDS trial may warrant LNA043’s entry into a phase III trial.

TPX-100 alleviated pain but did not improve the joint structure in a phase II clinical trial. Retrospective MRI analysis suggested a possible reduction in osteophytosis, but whether or not this finding will warrant a phase III trial is yet to be determined. Further investigation of TPX-100 in a longer phase III trial is necessary to determine its true potential as a DMOAD as an eight-week treatment period may be too short to produce long-term structural benefits.

#### 4.1.3. Gene Therapies

Gene therapies are a relatively new class of treatments for OA which utilize the transfer of genetic material to modulate gene expression. They have previously been associated with an increased risk of certain cancers, hypersensitivity reactions, and organ damage, but improved safety profiles have warranted an upward trend in FDA approvals over the last 5 years [65]. They remain one of the costliest drug classes, limiting their accessibility.

TG-C is the only gene/cell therapy for OA included in this review and is one of very few DMOADs to progress to a phase III trial. TG-C is also the only DMOAD included in this review which made statistically significant improvements in both joint pain and structure without major safety concerns. The careful regulation of TGF-β production is an important safety consideration as the overproduction of TGF-β is associated with multiple skeletal disorders [66,67]. While many gene therapies are permanent, the irradiation of TG-C donor cells induces quiescence and limits TGF-β production to a single cellular life cycle. Barring new safety concerns in ongoing phase III clinical trials and pending positive results, TG-C may be the first DMOAD to receive FDA approval.

Gene therapy is an exciting and rapidly growing avenue for DMOAD development, and several other novel agents have recently entered clinical trials in the wake of TG-C’s success. ICM-203 and FX201, two viral IA gene therapies, are currently being tested in phase I trials, and XT-150, a non-viral IA gene therapy, has surpassed phase I trials and completed two phase II trials [65], though the results have not yet been published on clinicaltrials.gov.

#### 4.1.4. Senolytic Agents

Senolytic agents selectively lyse the pro-inflammatory senescent cells which accumulate with age and aim to reduce the host inflammatory response associated with OA [68]. Senescent cells survive by upregulating antiapoptotic pathways, which can be transiently disabled to allow apoptosis to occur without affecting healthy cells. However, the removal of a cell line that is not fully understood carries inherent risk. Some studies indicate that senescent cells may have healing properties, including limb regeneration in newts [69] and cutaneous wound repair in mice [70]. While the removal of senescent chondrocytes has had positive effects in pre-clinical OA trials, additional human studies are needed to determine if promoting cell death could produce off-target effects.

The only senolytic agent which has completed a phase II clinical trial for OA is UBX0101. While a single dose of UBX0101 proved generally well tolerated and safe, it did not meet its primary endpoint of improving the WOMAC pain scores. Whether multiple treatments could safely improve its efficacy is yet to be determined. Senolytics are a promising future avenue for multiple age-related and chronic conditions but will remain clinically insignificant until clinical trials can demonstrate their efficacy.

#### 4.1.5. Monoclonal Antibodies

Anti-inflammatory monoclonal antibodies are highly specific agents commonly used in the treatment of rheumatoid arthritis. However, they can be expensive, carry a risk of immune reactions, and must be administered via an IV or SC injection. The only disease-modifying monoclonal antibody included in this review was Lutikizumab. Lutikizumab had no effect on the WOMAC pain scores or JSW and was associated with local inflammatory reactions in subjects. Anti-nerve growth factor (NGF) monoclonal antibodies—such as Tanezumab, Fulranumab, and Fasinumab—have also been tested in clinical trials but are classified as analgesic agents rather than disease-modifying. Though effective in alleviating symptoms, they too have been associated with safety concerns due to an increased risk of rapidly progressing OA [71,72].

Canakinumab is an anti-IL-1β antibody therapy that is FDA-approved for Still’s disease, periodic fever syndrome, and gout flares. It was tested in subjects with OA in a 2010 phase II clinical trial (NCT01160822), but, like Lutikizumab, failed to demonstrate significant efficacy. However, recent post hoc analyses from the CANTOS trial (NCT01327846) found that post-myocardial infarction patients treated with Canakinumab required significantly fewer joint replacement surgeries [73], suggesting that the inhibition of IL-1β may be chondroprotective.

### 4.2. Challenges to DMOAD Development

Although many DMOADs have shown promise in pre-clinical studies, a clinically meaningful response to treatment has remained elusive in human trials. OA is characterized by a highly complex immunologic landscape, the nuances of which are difficult to capture in cell-based and murine models. While a single-target agent may produce significant results in vitro, it is difficult to predict how said agent will perform when genetic influences and psychosocial factors are introduced. The general lack of success of novel DMOADs in human trials is multifaceted but can be attributed to three primary reasons: limitations in the trial design, disease heterogeneity, and off-target effects.

A lack of standardization among OA clinical trials creates challenges in interpreting and comparing results. The primary endpoint measures vary between trials, with pain scales and imaging being common choices. Pain is a hallmark symptom of OA but is highly subjective and susceptible to the placebo effect. It is also measured inconsistently, with some studies electing to use the WOMAC pain scale and others utilizing numerical ratings (scale of 1–10). When measured alone, pain provides an incomplete picture of clinical OA severity. Some patients may report changes in their physical activity level, mobility, or rescue medication use (i.e. NSAIDs, opioids) without noticing a significant difference in pain, and these parameters are not always measured. Imaging is more objective and commonly used to monitor and diagnose OA in the clinic, but the relationship between structural and clinical OA progression and pain is not always linear [74]. High-quality magnetic resonance imaging (MRI) can assess changes in the cartilage thickness, while X-rays can visualize osteophytes and JSN. However, the accurate reporting of such small measurements may be difficult and vary between radiologists.

Additional barriers to DMOAD development lie in the inherent nature of the disease. OA is highly complex and heterogenous, with the disease trajectory and response to treatment varying significantly from person to person [15]. There is currently no method to accurately predict the clinical prognosis, contributing to variable rates of disease progression in study populations. Most cases of OA progress slowly and are treated over decades, but a majority of clinical trials have primary endpoints between 6 and 24 months and relatively limited follow-up periods. The results of the CANTOS trial, which had a median follow-up time of 3.7 years, suggest that it may take several years for a DMOAD to exert any observable effects on joint pain or the structure. Longer trials would provide a better context for the evaluation of a drug’s efficacy but are time-consuming, costly, and burdened by large numbers of patients lost to follow-up, especially since OA primarily affects older individuals with multiple comorbidities. While many newer trials have begun expanding their follow-up periods, the possibility that a successful DMOAD candidate has already been prematurely overlooked should be considered.

Disease heterogeneity further complicates the selection of enrollment criteria in clinical trials. The inclusion criteria of most studies are constructed from some combination of age, BMI, baseline pain scores, and Kellgren–Lawrence (KL) structural imaging grades. At present, the majority of OA interventional trials include patients with KL grade 2 or 3 disease, often with a greater number of subjects falling into the grade 3 category. Grade 3 patients have late-stage structural disease and exhibit multiple osteophytes, significant JSN, sclerosis, and possibly bony deformities in imaging studies [75]. They often comprise the majority of study populations due to the presumption that they have the most to gain from DMOAD treatment. However, the predominant inclusion of KL grade 3 patients may be detrimental if these patients have surpassed a threshold beyond which disease progression can no longer be modified.

Finally, safety is of the highest importance in drug development, and most DMOADs have failed to demonstrate a favorable risk–benefit profile. As a non-fatal chronic disease, drugs to treat OA must have a particularly exceptional safety profile. A limitation of this review is the exclusion of phase I trials, as providing proof of safety is the primary objective at this stage. While only phase II and III clinical trials were included, there are many DMOADs which did not pass phase I due to safety concerns. Therefore, the prevalence of AEs in this review has been underestimated. Likewise, DMOAD trials conducted prior to 2010 were not included but still provide valuable lessons in DMOAD development. This review should not serve as a comprehensive overview of all DMOAD clinical trials but rather a focused discussion of DMOADs of recent or current interest. A final limitation of the findings discussed in this review is their potential lack of generalizability to non-European populations. The majority of included trials were conducted in countries with predominant European ancestry—including the US and Canada, European countries, and Australia—followed by Asian countries.

### 4.3. Recommendations for Future Research

Additional research is needed to address the challenges to DMOAD development. Improvements in the clinical trial design, the development of effective combination therapies, and the discovery of predictive biomarkers for the early detection, monitoring and prevention of OA could yield considerable advancements in this field.

#### 4.3.1. Improvements in Clinical Trial Design

In order to address these challenges, we must first define our target population. The assumption that patients with severe or rapidly progressing OA will experience greater outcomes lacks evidence and may be contributing to the failure of DMOADs in clinical trials. The target population in which DMOADs exhibit maximum efficacy may be patients with KL grade 1 or 2 OA, who demonstrate mild or moderate structural disease, respectively. Trials conducted in this population may yield greater outcomes because a greater level of physiologic joint function is intact. While patients with mild OA may have the most to gain from DMOADs, they may be unwilling to enroll in experimental trials if the risk of AEs outweighs their symptomatic burden, making recruitment an ethical challenge. The best population to include may be patients with moderate OA who are beginning to fail conservative treatment but have not yet progressed to severe, end-stage disease. Traditional joint replacement surgery may remain the best avenue for patients with end-stage OA.

There is increasing support for efforts to define OA phenotypes [76], which could improve patient stratification. The ability to characterize patients by various socio-demographic risk factors, clinical characteristics, and genetic profiles would allow investigators to determine their optimal treatment strategy. A recent reviewsubcategorized experimental DMOADs by their ability to target three distinct molecular OA endotypes—cartilage-driven, bone-driven, and inflammation-driven—to highlight the need for precision medicine techniques in OA management [77]. For example, Lorecivivint and Sprifermin may be most useful in treating patients with cartilage-driven OA, while MIV-711 may be most effective in the bone-driven endotype. The universal classification of OA by the phenotype and/or endotype and the adoption of this classification system in clinical trial enrollment and stratification methods may improve outcomes. To our knowledge, none of the DMOAD clinical trials conducted to date have utilized this strategy.

We must next develop clinical trials which prioritize clinical significance, rather than statistical significance. In order for a novel agent to be clinically useful, it should modify both symptomatic and structural OA progression. While several studies have acknowledged this, interventional trials continue to place a primary emphasis on either pain or structure, rather than considering both outcomes equally. For example, the FORWARD Sprifermin study found no statistically significant difference in pain reduction between the experimental vs. placebo groups but failed to adequately assess the agent’s symptomatic benefit. The investigators assert that the trial was designed to assess the structural modification, rather than symptoms, and note that a limitation of their study design was that they did not track the frequency of rescue medication use or changes in physical activity [20]. This is a missed opportunity for the FORWARD study because it leaves several questions regarding Sprifermin’s clinical benefit unanswered.

The standardization of endpoint measures and adoption of composite outcomes would improve the clarity and impact of clinical trial outcome reporting, the challenge here being that competing companies would have to agree. Trials should prioritize both clinical and structural outcome measures, including pain, mobility, physical activity, rescue medication use, overall wellbeing, and imaging-based measures such as JSW, cartilage thickness, and osteophytosis. The development of a composite score which combines these measurements would provide better insight into the clinical significance of an experimental DMOAD. A previous study found that a composite score including both pain and the activity level correlated better with the KL structural grade than did the traditional WOMAC pain score alone [78].

Finally, the continued expansion of trial follow-up periods will improve the observation of long-term safety and efficacy. The 10-year LNA043 trial, for example, will be far more informative of an agent’s lasting effects than trials lasting 1–2 years. Studies offering incentives and convenient methods of follow-up—including virtual visits or self-completed surveys—may increase patient retention. Prioritizing the inclusion of patients with mild-to-moderate, rather than severe, OA may also reduce attrition by lowering the mean age of enrollment and risk of mortality.

#### 4.3.2. Early Detection and Prevention of OA

While late-stage OA is easily diagnosed, a reliable way to detect the disease prior to the development of symptoms has not been established. Because the progression of OA is so unpredictable, the ability to treat the disease in its pre-clinical stages could offer a great benefit. The identification of a predictive biomarker for OA could allow for early intervention and favor drugs that are chondroprotective, rather than restorative. These biomarkers could be attained from a patient’s blood, urine, or synovial fluid and provide an individualized metabolomic profile. Recent studies have identified several molecules—such as metabolites, miRNAs, inflammatory cytokines, and collagen degradation products—which may predict the disease incidence, prognosis, and therapeutic efficacy [79,80,81]. However, a consensus on which biomarkers are strongly predictive of OA in a clinical setting has not yet been reached.

The accurate measurement of OA biomarkers could also transform DMOAD development and testing. The ability to quantify the disease severity and rate of progression could improve patient selection and study homogeneity. Investigators could preferentially enroll patients with a high risk of progression and reduce the enrollment of non-progressors or patients whose OA is stagnant. Biomarkers could also be used to enroll patients who are asymptomatic, but at a high risk of developing OA, in primary prevention trials. Finally, biomarkers may allow for early decision making in clinical trials if used as an endpoint measure. While clinical outcomes may take years to present, regular measurements of biomarkers may predict the DMOAD performance sooner. This could reduce the financial burden associated with longer OA trials and facilitate the termination of trials unlikely to benefit patients.

#### 4.3.3. Combination Therapies

An important consideration in DMOAD development is whether targeting a single aspect of a complex disease is capable of producing clinically significant results. A more realistic niche for DMOADs may be as adjuvant agents or combination therapies. Whether or not multiple agents could be delivered in tandem requires further investigation but may improve outcomes. An ideal combination therapy would provide both symptomatic and structural benefits. For example, Lorecivivint, an anti-catabolic drug which improves pain, could benefit from supplementation with an anabolic agent such as Sprifermin. DMOADs which improve the joint structure without reducing pain—such as MIV-711, Sprifermin, and LNA043—may also benefit from the concurrent use of first-line pharmacologic agents such as NSAIDs. However, there are both regulatory and dosage challenges in combination approaches.

Of course, lifestyle modifications such as proper nutrition, weight management, and physical therapy remain crucial facets of a comprehensive treatment plan. In their current state, DMOADs are unlikely to replace these first-line strategies but may be a useful adjunct to delay or prevent surgery when conservative treatment alone is insufficient. While the search for a novel agent to slow, prevent, or reverse OA progression is exciting, we should not forget that the tools to modify disease by promoting a healthy lifestyle are already at our disposal.

## 5. Conclusions

Success in DMOAD clinical trials has been scarce, with no novel agents demonstrating clinically significant efficacy. Of the eleven DMOADs included in this review, six have demonstrated statistically significant evidence of a structural or symptomatic benefit without major safety concerns: Lorecivivint, TG-C, Sprifermin, TPX-100, MIV-711, and LNA043. The remaining five compounds—SAR113945, GLPG1972, Cindunistat, UBX0101, and Lutikizumab—provided no benefit or were associated with AEs, indicating an unfavorable risk–benefit profile. TG-C was the only agent to provide both symptomatic and structural benefits. The pursuit of a successful DMOAD will require great flexibility and interdisciplinary teamwork to overcome significant challenges. Longer trials with refined inclusion criteria and endpoint measures, the development of effective biomarkers or screening methods, and combination therapies are key to moving forward in DMOAD development.

## Figures and Tables

**Figure 1 biomedicines-13-00355-f001:**
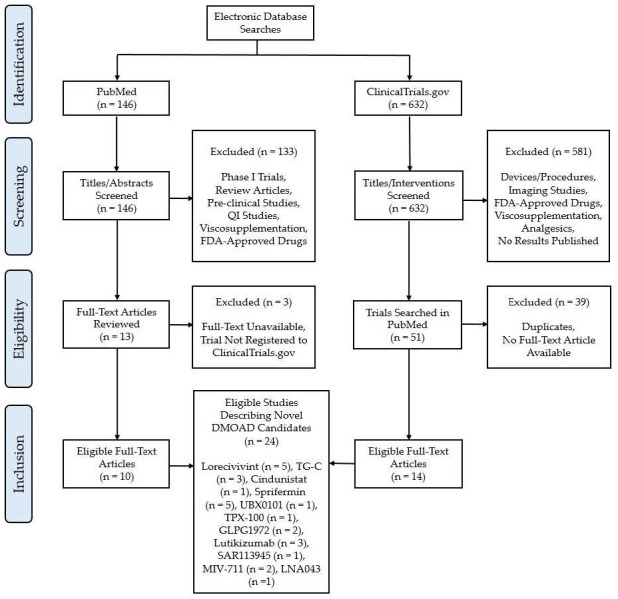
Flow diagram depicting search strategy, screening, and inclusion/exclusion criteria.

**Table 1 biomedicines-13-00355-t001:** Novel DMOAD candidates in phase III clinical trial (2010–2024).

Compound	Drug Class	Clinicaltrials.gov Identifier	Phase	Route, Frequency
Lorecivivint (SM04690)	Wnt/β-catenin inhibitor	NCT02095548	Phase I	IA, Single Injection
		NCT02536833	Phase IIa	
		NCT03122860	Phase IIb	
		NCT03928184	Phase III	
		NCT04385303	Phase III	
		NCT04520607	Phase III	
		NCT05603754	Phase III	
		NCT04598542	Phase III	
TissueGene-C	Pro-TGF-β gene therapy	NCT02341391	Phase I	IA, Single Injection
		NCT00599248	Phase I	
		NCT01671072	Phase II	
		NCT02341378	Phase II	
		NCT01825811	Phase II	
		NCT02072070	Phase II	
		NCT01221441	Phase III	
		NCT03203330	Phase III	
		NCT03291470	Phase III	
Cindunistat (SD-6010)	iNOS inhibitor	NCT00565812	Phase II	PO, Once Daily
		NCT01438918	Phase III	

Abbreviations: IA = intra-articular; PO = by mouth. Phase IIa = dosage and early efficacy, Phase IIb = efficacy at optimal dosage determined in Phase IIa.

**Table 2 biomedicines-13-00355-t002:** Novel DMOAD candidates in phase II clinical trial (2010–2024).

Compound	Drug Class	Clinicaltrials.gov Identifier	Phase	Route, Frequency
Sprifermin	FGF-18 analog	NCT01033994	Phase I	IA, Every 6 Months
		NCT01919164	Phase II	
UBX0101	Senolytic	NCT04229225	Phase I	IA, Single Injection
		NCT03513016	Phase I	
		NCT04129944	Phase II	
		NCT04349956	Phase II	
TPX-100	MEPE analog	NCT01925261	Phase II	IA, Once Weekly
GLPG1972/S201086	ADAMTS-5 inhibitor	NCT03311009	Phase I	PO, Once Daily
		NCT03595618	Phase II	
Lutikizumab (ABT-981)	IL-1*α*/*β* inhibitor	NCT01668511	Phase I	SC, Every 2 Weeks
		NCT02087904	Phase II	
SAR113945	IKK/NF-κB inhibitor	NCT01113333	Phase I	IA, Single Injection
		NCT01463488	Phase I	
		NCT01511549	Phase I	
		NCT01598415	Phase IIa	
MIV-711	Cathepsin K inhibitor	NCT02705625	Phase IIa	PO, Once Daily
		NCT03037489	Phase IIb	
LNA043	ANGPTL3 analog	NCT02491281	Phase I	IA, Once Weekly
		NCT04564053	Phase I	
		NCT03275064	Phase IIa	
		NCT04864392	Phase IIb	
		NCT04814368	Phase II	

Abbreviations: IA = intra-articular; SC = subcutaneous; PO = by mouth. Phase IIa = dosage and early efficacy, Phase IIb = efficacy at optimal dosage determined in Phase IIa.

## Data Availability

No new data were created or analyzed in this study. Data sharing is not applicable to this article.

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
