# Peer review of "Advances and Challenges in the Pursuit of Disease-Modifying Osteoarthritis Drugs: A Review of 2010–2024 Clinical Trials"

_biomedicines, 2025, doi:10.3390/biomedicines13020355_

Round 1

Reviewer 1 Report

Comments and Suggestions for Authors

1.  Despite the high disease burden of osteoarthritis (OA) and the development of various types of disease-modifying osteoarthritis drugs (DMOADs), there are almost no drugs that have undergone sufficient clinical validation to be used in real-world clinical practice. In this context, this article provides a well-organized review of recent clinical trials.

2.  The approach of discussing the need for DMOADs by explaining changes in the understanding of epidemiology and pathogenesis in the introduction, as well as the limitations of existing treatment medications, seems logical and natural.

3.  The method of reviewing each DMOAD candidate by first explaining the drug mechanism and then presenting the results of clinical trials is also helpful for understanding each drug in detail.

4.  The early part of the Discussion section classifies DMOAD candidates based on their mechanisms of action, while the latter part discusses clinical applications. Given that OA is increasingly understood as a heterogeneous disease with various endotypes and phenotypes rather than a single pathophysiological condition, it would be beneficial to mention if any DMOAD candidates have been studied based on endotype and/or phenotype.

Reviewer 2 Report

Comments and Suggestions for Authors

The manuscript titled "Advances and Challenges in the Pursuit of Disease-Modifying Osteoarthritis Drugs: A Review of 2010–2024 Clinical Trials" presents a comprehensive review of novel drugs that aim to modify the progression of osteoarthritis (OA). By analyzing clinical trials conducted between 2010 and 2024, the study identifies challenges and achievements in the development of disease-modifying osteoarthritis drugs (DMOADs). While the topic is highly relevant and timely, the manuscript suffers from several issues related to methodological rigor, clarity of arguments, and presentation of key findings.

1. While the manuscript provides an extensive overview of clinical trials, it fails to critically evaluate the reasons behind the lack of success for many DMOADs. For instance, issues such as variability in trial endpoints, the complex pathophysiology of OA, and challenges in patient stratification are mentioned but not explored in depth. Adding detailed analyses of these aspects would enhance the manuscript's contribution to the field.

2. The manuscript provides unequal attention to different classes of DMOADs. For example, while small molecule inhibitors and recombinant protein analogs are extensively discussed, other promising approaches like senolytic agents or gene therapies receive comparatively limited focus. Balancing the discussion would provide a more comprehensive perspective.

3. The conclusions about clinical relevance are overly general. For example, while the text acknowledges the limited structural benefits of certain agents like Sprifermin, it does not discuss what these limitations mean for future research or clinical adoption.

4. The manuscript touches on the importance of early detection through biomarkers but does not adequately explore how these could transform DMOAD development.

5. The focus on statistically significant findings often overshadows the discussion of clinical significance. For instance, while a 0.05 mm increase in cartilage thickness is statistically significant, its practical impact on patient outcomes is not discussed adequately.

6. Although the authors acknowledge some limitations, key issues like the generalizability of findings to non-European populations and the challenges of translating preclinical findings to clinical success are insufficiently addressed.

7. The recommendations for future research are generic and lack specificity. For example, the suggestion to extend trial durations could be supplemented with concrete strategies for addressing patient attrition or defining meaningful endpoints.

Round 2

Reviewer 2 Report

Comments and Suggestions for Authors

The author has made all the recommended and necessary corrections, and the text is now acceptable in its current form.